# ED-B-Containing Isoform of Fibronectin in Tumor Microenvironment of Thymomas: A Target for a Theragnostic Approach

**DOI:** 10.3390/cancers14112592

**Published:** 2022-05-24

**Authors:** Iacopo Petrini, Martina Sollini, Francesco Bartoli, Serena Barachini, Marina Montali, Eleonora Pardini, Irene Sofia Burzi, Paola Anna Erba

**Affiliations:** 1General Pathology, Department of Translational Research & New Technologies in Surgery and Medicine, University of Pisa and Azienda Ospedaliero Universitaria Pisana, 56100 Pisa, Italy; iacopo.petrini@unipi.it; 2Department of Biomedical Sciences, Humanitas University, Pieve Emanuele, 20090 Milan, Italy; martina.sollini@hunimed.eu; 3Diagnostic Imaging Department, IRCCS Humanitas Research Hospital, Rozzano, 20089 Milan, Italy; 4Regional Center of Nuclear Medicine, Department of Translational Research and New Technology in Medicine, University of Pisa and Azienda Ospedaliero Universitaria Pisana, 56126 Pisa, Italy; francesco.bartoli@med.unipi.it; 5Laboratory of Hematology, Department of Clinical and Experimental Medicine, University of Pisa, 56126 Pisa, Italy; serena.barachini@unipi.it (S.B.); marina.montali@unipi.it (M.M.); eleonora.pardini@med.unipi.it (E.P.); i.burzi@studenti.unipi.it (I.S.B.); 6Department of Nuclear Medicine and Molecular Imaging, Medical Imaging Centre, University Medical Center Groningen, 9700 RB Groningen, The Netherlands

**Keywords:** thymic epithelial tumors, tumor microenvironment, fibronectin, target therapy, theragnostic

## Abstract

**Simple Summary:**

The extra-domain B fibronectin (ED-B FN) is highly expressed in thymic epithelial tumors (TETs), as demonstrated by in vivo targeting using 131I-labeled L19 small immunoprotein (131I-L19-SIP) and immunohistochemistry with a predominant expression by stromal cells of a thymoma microenvironment rather than epithelial cells. Such high expression derived from the induction of stromal cells shifts FN production to the ED-B subtype. Our results suggest that Radretumab radioimmunotherapy (R-RIT) inefficacy is not related to low TET ED-B expression but to multifactorial aspects including patients’ inherent characteristics, the pattern expression of the target, the biological characteristics of the tumor, and the format of the target agent, which contribute to the resistance of tumor cells to treatment.

**Abstract:**

Aim: to exploit tissue-specific interactions among thymic epithelial tumor (TETs) cells and extra-domain B fibronectin (ED-B FN). Material and methods: The stromal pattern of ED-B FN expression was investigated through tumor specimen collection and molecular profiling in 11 patients with recurrent TETs enrolled in prospective theragnostic phase I/II trials with Radretumab, an ED-B FN specific recombinant human antibody. Radretumab radioimmunotherapy (R-RIT) was offered to patients who exhibited the target expression. Experiments included immunochemical analysis (ICH), cell cultures, immunophenotypic analysis, Western blot, slot-blot assay, and quantitative RT-PCR of two primary thymoma cultures we obtained from patients’ samples and in the Ty82 cell line. Results: The in vivo scintigraphic demonstration of ED-B FN expression resulted in R-RIT eligibility in 8/11 patients, of which seven were treated. The best observed response was disease stabilization (*n =* 5/7) with a duration of 4.3 months (range 3–5 months). IHC data confirmed high ED-B FN expression in the peripherical microenvironment rather than in the center of the tumor, which was more abundant in B3 thymomas. Further, there was a predominant expression of ED-B FN by the stromal cells of the thymoma microenvironment rather than the epithelial cells. Conclusions: Our data support the hypothesis that thymomas induce stromal cells to shift FN production to the ED-B subtype, likely representing a favorable hallmark for tumor progression and metastasis. Collectively, results derived from clinical experience and molecular insights of the in vitro experiments suggested that R-RIT inefficacy is unlikely related to low target expression in TET, being the mechanism of R-RIT resistance eventually related to patients’ susceptibility (i.e., inherent characteristics), the pattern expression of the target (i.e., at periphery), the biological characteristics of the tumor (i.e., aggressive and resistant phenotypes), and/or to format of the target agent (i.e., 131I-L19-SIP).

## 1. Introduction

Thymic epithelial tumors (TETs) are rare malignant neoplasms with an annual incidence of approximately 0.32 every 100,000 people/year [1]. According to the 2020 classification of the World Health Organization, TETs are divided into thymomas and thymic carcinomas. Thymomas account for about 90% of TETs and are further classified in A, AB, B1, B2, and B3 histotypes depending on their morphological features [2]. About 10–15% of patients experience recurrence after 5 years (range 3–7 years) [3,4] from the primary treatment with curative intent [3,4,5,6]. In recurrent disease, regardless of histology, chemotherapy-based regimens become much less effective. Given such poor response of TETs to chemotherapeutic agents, other treatment options have been investigated, such as target agents and immunotherapy. Multikinase inhibitors sunitinib and lenvatinib have been tested with promising results (objective response rate of 26% and 38%, respectively) [7,8,9]. Immunotherapy, although effective, is currently reserved to selected cases due to the increased risk of auto-immune-mediated complications, especially in thymomas [10,11].

In recent years, tumor microenvironments (TME) have revealed novel targets, becoming an attractive field of research in drug development. The literature is rich with examples of identification of TETs’ TME immune components and their respective correlation with tumor molecular subtypes [12,13,14]. Despite being strongly involved in supporting cancer growth and the outcome of therapeutic approaches, the extracellular matrix (ECM) is probably the component of the TME that initially received the least attention [15]. One of the most interesting stromal targets is fibronectin (FN), a stably and abundantly expressed protein on newly formed blood vessels of cancer tissues, theoretically undetectable in adult healthy tissues except in specific conditions including tissues remodeling and repair, fibrosis, and cell migration [16,17,18]. Using Radretumab (131I-L19SIP), a fully human armed antibody in the small immunoprotein (SIP) format that specifically binds with high affinity the extra-domain B (ED-B) splice variant of the ECM FN, we obtained encouraging results in both the preclinical setting [18,19] and clinical trials. EDB-target therapy has been explored in solid tumors and hematological malignancies using either Radretumab (NCT01242943; NCT01124812; NCT01125085) or antibody–cytokine fusion proteins (L19-IL2, Darleukin and L19-TNF, Fibromun) on the tumor site by homing to the EDB-FN. Despite an excellent in vivo demonstration of the targeting, treatment with Radretumab had efficacious results only in anecdotal cases. One of the major obstacles in developing a TME-specific treatment strategy is the presence of tumor heterogeneity, which is also constantly evolving throughout the disease’s development and in response to therapy, multiple compensatory mechanisms and feedback loops being the basis for therapy evasion and the development of resistance. Collectively, so far, targeted therapies have not yielded the expected results in recurrent/refractory TETs, suggesting that the success of clinical trials in this context requires greater attention to tumor specimen collection and to molecular profiling. Therefore, to exploit tissue-specific interactions among TET cells and ED-B FN, potentially enlightening the mechanisms sustaining TET radioresistance, we investigated through tumor specimen collection and molecular profiling the stromal pattern of ED-B FN expression in patients with recurrent TETs treated with Radretumab.

## 2. Materials and Methods

### 2.1. Clinical Experience

Among all subjects enrolled at the University of Pisa into the prospective phase I/II trials EudraCT no. 2005-000545-11 (approved by the Ethics Committee of the Azienda Ospedaliero-Universitaria Pisana on 14 September 2006 and emended on 2 May 2007) and EudraCT no. 2007-007241-12 (Ethics Committee of the Azienda Ospedaliero-Universitaria Pisana approval number 2622/2008) sponsored by Philogen SpA (Siena, Italy), we recruited 11 patients with recurrent TETs. The studies, approved by national authorities and local ethics committees, were conducted in accordance with the European regulations, the Declaration of Helsinki, and the International Conference on Harmonization Good Clinical Practice Guidelines. All patients individually dated and signed the informed consent form before being enrolled. Specific details about the clinical trial including Radretumab radiolabeling, imaging acquisition protocol, dosimetric estimates, and treatment schedules have been previously published [20]. Briefly, all enrolled patients were selected for treatment after a diagnostic phase consisting of sequential scintigraphic images and blood samples collected at different time points (24, 48, 72, 96 h, and 8 days) after the injection of a diagnostic activity of Radretumab (185 MBq). The aim of the diagnostic phase was to demonstrate the in vivo expression of the target (i.e., expression of visual assessment by comparing uptake in the target to background) and to estimate the adsorbed doses to target lesions(s), normal and dose-limiting organs (i.e., bone marrow), respectively. Accordingly, Radretumab radioimmunotherapy (R-RIT) was offered to patients who met a priori defined criterion based on the target/non-target adsorbed dose ratio. Patients who did not fulfil the target/non-target adsorbed dose ratio criterion ended the protocol. Treatment was administered 14–30 days after the dosimetric phase, injecting a fixed activity of Radretumab in phase I (3.7 or 5.55 GBq EudraCT study no. 2005-000545) or according to the recruitment cohort in the dose-escalation protocol (4.1 GBq/m^2^, 5.1 GBq/m^2^, and 6.2 GBq/m^2^, respectively—EudraCT no. 2007-007241-12). All patients were imaged about 8 days after R-RIT to confirm the targeting. Adverse events (AEs) and drug-related toxicity were recorded as per study protocol for at least 30 days after the last Radretumab administration (diagnostic or therapeutic) and graded according to the common terminology criteria adverse events (CTCAE v.3). Treatment response was assessed clinically and radiologically. Clinical response assessment evaluated symptoms and lab tests. RECIST criteria version 1.1 [21] were used for radiological treatment response. Repeated R-RIT (at least 3 months apart) was allowed based on clinical judgement, dosimetric estimates, safety, treatment response, and clinical follow-up. Baseline patients’ characteristics, including previous treatments, are summarized in Appendix A.

### 2.2. Immunohistochemistry

Paraffin-embedded samples of thymoma were obtained from the Pathology department of our university. Thymic epithelial tumors were classified according to the 2004 version of the WHO classification of thoracic tumors [22]. FFPE samples were cut into 4 μm slices, deparaffinized with xylene, and rehydrated in graded ethanol.

Antigen retrieval was performed using steam pressure boiling in 2 mM EDTA, pH 8.0 for five minutes. Immunohistochemistry was performed using L19-IL2 fusion protein that specifically binds to ED-B FN such as previously reported [16,23] and the alkaline phosphatase anti-alkaline phosphatase (APAP) method was adopted to reveal the staining. For each sample, 10 microscopic fields were evaluated for the ED-B FN sating at 400× *g* magnification using an Olimpus AX70 microscope equipped with a zoom ocular and PlanAPO objectives.

### 2.3. Cell Cultures

The DU-145 human prostate cancer and the MEWO malignant melanoma cell lines were obtained from ATCC (American Type Culture Collection, Washington, DC, USA). Ty82 human thymic carcinoma cells were purchased from Japanese Collection of Research Bioresources Cell Bank (Ibaraki, Japan). The cells were cultured in a 5% CO_2_ incubator at 37 °C with RPMI-1640 medium (Gibco, 11835-063, Waltham, MA, USA) supplemented with 10% fetal bovine serum (Gibco, 16000-044, USA), 100 U/mL penicillin, and 100 μg/mL streptomycin (Gibco, 15140-122, USA).

Two fresh samples of TET, named AMt and NF, were obtained from the resection of TETs during surgery at the University of Pisa. Tumor tissue was mechanically dissociated and cultured in a 6 cm dish and then seeded on hydrophilic plastic in KGM-Gold™ Growth Medium (Lonza Group Ltd., Basel, Switzerland) supplemented with 1% Glutamax^®^, 1% penicillin–streptomycin (Life Technologies, Carlsbad, CA, USA). After confluence, cells were harvested by TryPLE Select^®^ (Life Technologies) digestion and then replaced and cultured to next passage at a 1:3 ratio.

### 2.4. Immunophenotypic Analysis

AMt and NF cells, at second, fifth, and eighth passage, were harvested, and a total of 5 × 10^5^ cells from single-cell suspensions were dispensed per each tube. Samples were incubated for 30 min at 4 °C with labeled monoclonal antibodies (mAbs) specific for EpCAM-APC and CD90-FITC (Miltenyi Biotech, Bergisch Gladbach, Germany). Then, samples were washed twice and resuspended in MACSQuant™ Running Buffer (Miltenyi Biotech, Bergisch Gladbach, Germany). The flow cytometer was set using cells stained with isotype-identical antibody controls. Cells were gated on a forward (FSC) versus side scatter (SSC) plot in order to eliminate debris. Acquisition was performed collecting 10,000 events that were analyzed by MACSQuant^®^ Flow Cytometer using the MACSQuantify^®^ Software (Miltenyi Biotech, Bergisch Gladbach, Germany).

### 2.5. Western Blot

AMt, NF, DU145, and MEWO cells were lysed on ice using RIPA lysis buffer kit containing protease inhibitor cocktail (Santa Cruz Biotechnology Inc., Heidelberg, Germany). Total protein concentration was determined by BCA protein assay kit (Pierce, Thermo Scientific, Rockford, IL, USA). Equal amounts (30 μg) of protein were loaded and separated on precast Miniprotean Gel 4–20% (Biorad, Segrate, Milan, Italy). The separated proteins were transferred to nitrocellulose by Semi-Dry Trans-Blot Turbo System (Biorad); after blocking in TBS-T containing 5% non-fat dry milk (Biorad) for 2 h at room temperature, the membranes were incubated with primary antibody overnight at 4 °C. The primary antibodies utilized were anti-EpCAM (1:1000) and anti-Pan cytokeratin (1:300) (Abcam, Cambridge, UK), Rhodamine anti-β-Actin (BioRad, Hercules, CA, USA, 1:2500), and anti-Vimentin (Cell Signaling Technology, Danvers, MA, USA, 1:1000). Subsequently, the membranes were incubated with the appropriate HRP-conjugated secondary antibody (Bio-Rad, Hercules, CA, USA) for 1 h at room temperature. Protein bands were visualized using ECL detection reagent (Amersham, Glattbrugg, Switzerland) and a ChemiDoc Imaging System (Bio-Rad, Hercules, CA, USA). Image Lab software (Bio-Rad, Hercules, CA, USA) was utilized to quantify the density of each band by densitometric analysis. The relative expression of each protein was normalized to -Actin.

### 2.6. Slot-Blot Assay

Briefly, 60 μg of NF and Ty-82 cell lysate was spotted onto the nitrocellulose membrane at the center of the grid of a Bio-Dot SF Microfiltration Apparatus (BioRad, Milan, Italy). FN was used as a positive control. The membranes were dried and blocked by soaking in 5% BSA in TBS-T for 1 h at room temperature. The membranes were incubated for 1 h at room temperature with monoclonal antibodies anti-fibronectin BC-1 (1:500, Abcam, Cambridge, UK), L19-sip (1 µg/mL, Philogen SpA, Siena, Italy), and anti-β-Actin (Cell Signaling Technology, Danvers, MA, USA, 1:1000) diluted in BSA/TBS-T. After three washings with TBS-T, the membranes were incubated with secondary antibody HRP-conjugated goat anti-mouse IgG (BioRad, Milan, Italy, 1:1000) for 30 min at RT. Subsequently, the membranes were washed three times with TBS-T to remove unbound antibody and then incubated with Clarity Max ECL Substrate (BioRad, Milan, Italy) for 1 min. Chemiluminescence was detected with ChemiDoc MP Imaging System.

### 2.7. Quantitative RT-PCR

RNA extraction from AMt, NF, and Ty-82 was performed using RNeasy Mini Kit (Qiagen GmbH, Hilden, Germany), according to the manufacturer’s instructions. One µg of each RNA sample was reverse transcribed to cDNA by QuantiTect Reverse Transcription Kit (Qiagen GmbH), and 30-fold dilutions of cDNAs were analyzed by quantitative RT-PCR on iCycler-iQ5 Optical System (Bio-Rad Laboratories, Hercules, CA, USA), using SsoAdvanced SYBR Green SuperMix (Bio-Rad Laboratories, Hercules, CA, USA), running each sample in duplicate. Primers were designed from coding sequences published on Gene Bank database with the support of Beacon Designer v.7 Software (Premier Biosoft International, Palo Alto, CA, USA) (sequences are available upon request). Relative quantitative analysis was performed following 2^−ΔΔCt^ Livak method [24], and the geometric mean of four housekeeping genes (ACTB, ATP5B, GAPDH, HPRT) based on the GeNorm study [25] was used to normalize the ED-B FN mRNA expression.

## 3. Results

### 3.1. Clinical Experience

Table 1 summarizes the main results for each patient. Selective uptake of Radretumab (from faint to intense) was observed in target lesions 24–48 after the diagnostic administration for 10/11 TET patients (91%). In the remaining case, all lesions (liver metastases) were “cold”. Figure 1 shows different scintigraphic patterns of Radretumab uptake.

Normal organs and healthy tissues did not show any significant Radretumab uptake, with the only exception of thyroid in case of inadequate block. The calculated dosimetric estimates for target lesion(s) and bone marrow are summarized in Appendix A. In 8/11 patients, the estimated absorbed dose of beta radiation by the red bone marrow was below the TD5/5 of 2.5 Gy. The average estimated absorbed doses of beta radiation to healthy organs for all patients are summarized in Appendix A. 

Eight out of eleven patients met the target/non-target ratio criterion. Accordingly, seven patients were treated with Radretumab (mean administered activity 6.2 ± 2.7 GBq, range 2.96–9.435). One patient eligible for R-RIT was not treated because she withdrew consent (*n =* 1). R-RIT was repeated in three patients (mean administered activity in 7.6 ± 2.4 GBq, range 5.549–10.027). Images acquired after R-RIT—regardless of the first or the following(s), in case of repeated administrations—agreed with pre-treatment morphological imaging (i.e., concordance between the number and the site of lesions) and confirmed the antigen targeting in all cases. All patients who received the diagnostic (*n =* 11) and therapeutic administration (*n =* 7) of Radretumab were evaluable for the safety analysis. Overall, Radretumab was well tolerated, and repeated administrations were feasible.

After the diagnostic administration of Radretumab, 7/11 patients (64%) experienced adverse events, defined as serious (SAE) in two cases. SAE included grade 2 dyspnea and grade 3 metabolic/laboratory test alteration (transitory grade 3 increase in creatinine level associated with increased LDH and uremic acid values). Both SAEs were registered in patients eligible for R-RIT and considered disease-related (i.e., unrelated to the study drug). Not-serious AE included a grade 1 nausea (*n =* 1), transient worsening of pre-existing asthenia associated with articular pain (*n =* 1), and grade 2 hypersensitivity to perchlorate potassium (*n =* 1), after the diagnostic administration of Radretumab. None of the reported AEs were considered treatment related. Toxicity after R-RIT administration was, as expected, mainly hematological (see Table 1). Five patients (71%) experienced treatment-related hematological AEs ≥ grade 3, which included uncomplicated grade 3 thrombocytopenia (*n =* 1), grade 3 leucopoenia (*n =* 1, after the third R-RIT administration), and grade 3–4 lymphocytopenia (*n =* 3, one after the second R-RIT administration). Median time to the platelet nadir (<100 × 10^9^/L) was 32 days (range 17–48 days).

The most frequent non-hematological AEs after R-RIT included hypothyroidism (*n =* 3), grade 1/2 asthenia (*n =* 2), nausea (*n =* 1), and grade 2 cutaneous rush (*n =* 1). Patients who developed permanent hypothyroidism did not perform a thyroid block as required by protocol procedure (two patients had an allergic reaction to the first administration of Lugol’s solution, and one patient refused to assume it).

One patient died before treatment response assessment. Death, unrelated to R-RIT, was due to rapid progression of thymoma. All treated patients were included in the efficacy analysis, as detailed in Table 1. Six out of seven treated patients (86%) had symptoms relief demonstrated by both physical examination and medical records (decreased in analgesic drugs consumption, maintenance of physical activity), which combined result in improvement of patients’ quality of life at the cost of relatively low and easily manageable side effects. After the first R-RIT administration, five patients had stable disease, and one patient experienced disease progression according to RECIST criteria. One patient who received repeated R-RIT experienced a partial response with a reduction of approximately 50% of initial tumor burden after the second administration, obtaining a long-lasting disease control (Figure 2).

### 3.2. Immunohistochemical Analysis

Thymomas strongly expressed ED-B FN according to L19-IL2 staining. Interestingly, the staining was around neoplastic nodes (Figure 3), suggesting that ED-B FN is expressed in the peripherical microenvironment rather than in the center of the tumor. These results support the idea that predominantly stromal cells produce ED-B FN.

The ED-B FN antigen was strongly expressed in B3 thymomas, confirming the high specificity of the targeting demonstrated by scintigraphic images (Figure 3).

### 3.3. Primary Cell Cultures and Expression of ED-B Fibronectin

We established two primary cultures from two samples of thymoma collected during surgery: a type A (AMt) and a type B2 (NF) thymoma. Using flow cytometry, we observed that the cell culture obtained from type A thymoma expressed thymocyte-mesenchymal (CD90^+^) and epithelial (CD326^+^) markers at passage 1, but the epithelial marker CD326 was loose at passage 5. In the B2 thymoma, only CD90 was found since the first passage (Figure 4A,B). The growth of thymic tumor epithelial cells in vitro is difficult, and our observations confirm the outgrowth of stromal cells in both primary cultures. Moreover, the cultured cells had a fibroblast-like appearance when observed with an inverted phase microscopy (Appendix A).

Using Western blot, we evaluated the expression of EpCAM, vimentin, and pan-cytokeratin in primary cells after passages 9 and 5 in AMt and NF cells, respectively (Figure 4C, Appendix A). We included lysates of the melanoma MEWO and prostatic cancer DU145 cell lines as positive controls for vimentin and cytokeratin expression. EpCAM was not expressed in any of the cells evaluated, whereas vimentin was strongly expressed in NF and AMt cells. Pan-cytokeratin sating suggests the expression of type I cytokeratins in NF primary culture but not in AMt cells. The expression of vimentin has been described in epithelial cells of type A and AB thymomas [26], but the absence of pan cytokeratins excluded the presence of epithelial cancer cells in the AMt primary culture. On the contrary, B2 thymoma cells are vimentin-negative; therefore, stromal cells are present in the NF primary culture even if a subset of cytokeratin-positive cancer cells is still present. Normalized densitometric analysis of protein expression confirmed our observations (Appendix A). Western blot results confirmed the presence of tumor-associated stromal cells in both cultures, possibly with a sub-population of tumor epithelial cells.

Using reverse transcription PCR, we evaluated the expression of ED-B FN mRNA in the two primary cell cultures of thymoma and in the TY82 cell line. TY82 is a cell line derived from a NUT rearranged undifferentiated thymic carcinoma [27]. Interestingly, both AMt and NF cells expressed comparable levels of ED-B FN (average DeltaCt 1.575 SD 0.237 and 1.000 SD 0.077), whereas TY82 cells expressed much lower levels of mRNA (average DeltaCt 0.001 SD 0). The difference in mRNA expression of ED-B FN between primary cultured thymoma cells and the TY-82 thymoma cell line was statistically significant (*p* < 0.001) (Figure 4D).

Slot blot of cell lysates from NF cells at passages 9 and 10 showed the reactivity to L19-SIP and to the BC 1 antibody: a monoclonal antibody able to recognize ED-B^+^ FN isoforms. Commercially available fibronectin (i.e., a mix of fibronectin types) and TY-82 lysate were used as a positive control. ED-B fibronectin expression did not modify at the evaluated passages in NF cells. ED-B staining was more intense in the NF primary culture than in TY-82 cell lines, according to the BC-1 antibody. This assay confirmed that cells produced ED-B FN, being both BC 1 and L19-SIP-specific for ED-B FN (Figure 4E,F).

## 4. Discussion

Selective uptake of Radretumab was observed in all the target lesions of TET patients included in the present analysis, with the only exception being a case of liver metastases. Dosimetry confirmed R-RIT eligibility in a high percentage of cases (73%), similarly to data previously reported in lymphoma [20]. Although these findings confirmed in vivo stromal expression of ED-B FN, we observed poorer and shorted objective responses as compared to what we reported in lymphoma [20]. Nevertheless, it is known that different factors affect radiosensitivity [27], and lymphomas are listed among the most radiosensitive tumors [28]. The best recorded response in TET patients was disease stabilization (*n =* 5/7, 71%), which further improved to partial response after the second R-RIT administration in one case (#9). The mean duration of treatment response in the TET cohort was 4.3 months (range 3–5 months) as compared to 7.4 months (range 1–15 months) in lymphoma (with a mean of 5 months, range 3–6 months, when considering only patients experiencing disease stabilization).

Notably, R-RIT positively impacted on symptoms in both TETs and lymphoma patients, leading to decreased intake of concomitant medication and finally improving quality of life.

As for safety, R-RIT resulted in transitory and self-limiting hematological toxicity, as typical in RIT, even when the provisional dose to bone marrow was above the recommended safety threshold (TD 5/5 = 2.5 Gy). These data confirm that the bone marrow toxicity is not predictable based on dosimetric estimates, requiring an estimation of the bone marrow reserve through multiple parameters such as previous treatment regimens, disease burden and location, as well as other individual factors. As side effects, inadequate thyroid block was associated with the development of hypothyroidism.

Despite the demonstration of the target expression with imaging the R-RIT objective response, our cohort of TETs turned out to be very disappointing. Speculations about the lack of therapeutic efficacy include bias in patients’ selection, additional mechanisms besides in vivo demonstration of target expression, insufficient administrated activity, and radioresistance. First, heterogeneity of the disease in terms of biology, genetic characteristics, tumor stage, and burden should be considered. In particular, thymoma subtypes are biologically different, and they are not a continuum of diseases. The histological subtypes, as defined by the WHO classification, are strongly associated with multiple aberrations occurring at different levels, which defined specific genomic hallmarks [29]. Indeed, some of these differences can account for the different radiosensitivity of tumor cells and patients [27,30]. Further, TETs might present slow and indolent growth or rapid disease progress and metastasizes [7]. Hematogenous or lymphatic metastases are common in thymic carcinomas, thymomas frequently grow through local infiltration, and their most frequent way of diffusion is through pleural metastases. Therefore, the interpretation of the clinical results of the treatment as disease stabilization in single arm trials is non-univocal.

Insufficient effective dose to all the lesions should also be considered as a possible reason for low treatment efficacy. Radiation induced immune and stromal effects [31]. In animal models of thymoma and breast cancer, dendritic cells have been shown to be important in the antitumor T-cell-mediated immune response following radiotherapy through radiation-induced release of high-mobility group box 1 (HMGB1) by dying tumor cells, which act on toll-like receptor 4 (TLR4) expressed by dendritic cells, aiding in processing and cross-presentation of tumor-associated antigens to CD8^+^ T cells. Interestingly, the ED-A FN agonizes TLR4, and recently the immunological function of the ED-A containing an FN splice variant in vivo has been elucidated [32]. However, evidence has been reported suggesting that high-dose radiation regimens are required for optimal induction of T-cell immune responses following radiotherapy [31]. Nevertheless, dosimetric estimates in our cohort did not support this hypothesis since patients with stable disease (#2, #6, #8) had an estimated dose to target lesion up to 10 times higher than the responder patient (#9). In addition, the best objective responses in terms of size reduction according to RECIST were observed in patients with multiple measurable target lesions, apparently rejecting a direct link between number/size of the lesions and R-RIT effectiveness. Moreover, a negative correlation between radiation-induced lymphopenia and outcome has been reported in solid tumors [33,34]. In our cohort, the patient who exhibited the best response experienced a severe (G3) but transient lymphopenia. However, the limited sample size and the small number of events prevent further speculations. Heterogeneity of the target expression also warrants special consideration. We can advocate several limitations of the technique employed for the in vivo demonstration of the targeting expression. However, if the low activity administered in case of diagnostic scan could be considered as a cause for disease underestimation, post R-RIT scintigraphy obtained 7 days after treatment confirmed the same pattern of lesion locations and target expression. Further, strong stromal expression of ED-B FN was also confirmed in all patients with immunohistochemistry. However, with immunohistochemistry we found a specific pattern of ED-B FN in TETs at the periphery of the tumors. This specific pattern could be a consequence of its production by either cancer-associated fibroblasts (CAF) [35] or by a direct production from cancer cells during epithelial-to-mesenchymal transition (EMT) at the invasive periphery of the tumor, a process used to enhance their invasive and mobility potential [36,37]. Indeed, differently from other tumor types (e.g., melanoma), in TETs the overexpression of the ED-B isoform seems to be related to the acquisition of more aggressive phenotypes [37]. Together, the peripheral expressions of the target and the more aggressive behavior of the tumor could contribute to a scarce efficacy of R-RIT. Mizuno et al. [38] described different patterns of fibronectin and laminin distribution based on TET behaviors. Specifically, they reported a diffusely or partially intricate network surrounding tumor cells composed of fibronectin and laminin in TETs with low invasive potential. Conversely, more invasive tumors presented fibers containing fibronectin and laminin in the septa, blood vessels, and perivascular space [38]. These results suggested a correlation between the pattern of stromal components and tumor invasiveness, further supporting our hypothesis.

Therefore, the pattern distribution of the target is extremely critical when the candidate therapeutic agent is not designed to specifically target cancer cells as in the case of Radretumab. Moreover, the effects of radiation on different stromal constituents of tumors, investigated as secondary effects of external beam radiotherapy, depend on physical parameters such as total radiation dose, fraction size, fraction intervals, or number of fractions. Further, the type of the target could also influence treatment response, and although a similar effect on cellular and stromal agents could be hypothesized, it is likely that there exist significant differences in the two scenarios. Increasing our understanding of radionuclide target therapy radiobiology is critical to fully capitalize on the potential benefits of TME-targeted therapies. For instance, better characterization and modeling of the radiobiological tested agent, alone or in combination regimens, must be understood before we can optimize the dosing schedule. In this study, we generated two primary cultures from two thymomas. The cultures were a mixture of neoplastic and stromal cells. With subsequent passages, the stromal compartment of the culture had overgrown the neoplastic thymoma cells. This was expected because thymoma cells are difficult to grow in vitro. Indeed, only a few cell lines of TETs are available to date. Our data showed that stromal cells from TET samples expressed significant amounts of ED-B FN, confirming in vivo findings. Moreover, data on the TY-82 cell line confirmed the much lower expression of ED-B FN.

Such differences in terms of ED-B isoform expression between TY-82 cell line (pure epithelial cells) and the two primary cultures derived from recurrent TET patients (aggressive phenotypes), together with literature data—it has been recently shown that thymoma cells may undergo EMT [39], and we had showed that epithelial-to-mesenchymal transition (EMT) induces ED-B FN expression [37]—further supported the speculation that the overexpression of ED-B FN in TETs is related to EMT and the acquisition of more aggressive phenotypes. Indeed, our previous findings showed EB-FN upregulation in both malignant and tissue stem cells after induction of EMT, establishing EDB-FN as a potential biomarker for such a pro-metastatic process and a strong activator of the EMT programming, thus enabling acquisition of a metastatic phenotype [37].

Finally, mechanisms underlying treatment resistance should be considered. Several studies have showed how cells’ exposure to an FN-rich microenvironment stimulates a diverse set of behaviors related with tumor invasion, metastasis, proliferation, and resistance to pro-apoptotic signals. A recent study demonstrated that lung cancer cells cultured with fibronectin showed: increasing Erk and Rho pathways activity; activation of protein tyrosine kinase and inhibition of caspase 3; a decrease of cell cycle inhibitor p21 expression; and cell cycle promoter cyclin D1 stimulation. All of these factors allow cancer cells to ignore the pro-apoptotic signals, promoting proliferation pathways and enabling the development of tumor cells’ resistance to different drugs [40,41]. Furthermore, in an FN-rich microenvironment, adhesion to FN causes upregulation of pro-survival genes and inhibition of pro-apoptotic factors, with a subsequent increase in drug-resistance and radioresistance [42]. High FN was previously implicated in a highly differentiation-resistant oral epithelial cell line that displayed changes in morphology and gene expression indicative of EMT [43]. These findings highlight the role of adhesion signaling in EMT and progression to a radioresistant phenotype [44]. Jerhammar et al. confirmed the involvement of developmental processes and adhesion molecules in radioresistance and proposed FN expression as a biomarker of poor response to radiotherapy [45]. Taken all together, our data support the hypothesis that thymomas induce stromal cells to shift FN production to the ED-B type, this being a favorable hallmark for tumor progression and metastasis. Collectively, results derived from clinical experience and insights provided by in vitro experiments suggested that the cause of R-RIT inefficacy is unlikely related to expression of the target in TET, being the mechanism of R-RIT resistance eventually related to patients’ susceptibility (i.e., inherent characteristics), the pattern expression of the target (i.e., at periphery), the biological characteristics of the tumor (i.e., aggressive and resistant phenotypes), and/or to format the target agent (i.e., 131I-L19-SIP). For example, very recent data on the comparative analysis of two chemically defined antibody−drug conjugates and small molecule−drug conjugates products directed against the same molecular target have provided the biological proof of the suboptimal tumor uptake, limited to perivascular cancer cells of the antibody−drug conjugate molecules. On the contrary, small organic ligand already exhibited a homogeneous uptake in the neoplastic mass after 1 h, potentially overcoming such a limitation [46]. Similarly, the efficacy of ED-B chimeric antigen receptor (CAR) T-cell therapy has been recently reported in both in vitro experiments and in animal models [47]. Further, although β-emitting isotopes still represent the most extensively, clinically used agents, there is growing interest in the use of α-and Auger emitters [48], which will result in differences in radiobiology and therefore biologically effective doses.

Therefore, we believe that therapies with TME components including the one targeting ED-B FN with the opportunity for combination therapy remain of great interest in TET, for which molecular targets in cancer cells remain elusive even after large genomic sequencing evaluations [29]. Increasing evidence in molecular biology and gene expression profiles suggest that “It’s more important to know what sort of person a disease has than to know what sort of disease a person has” (Hippocrates http://www.brainyquote.com/quotes/authors/h/hippocrates.html, accessed on 20 May 2022). 

## 5. Conclusions

Our data showed a scarce efficacy of R-RIT in TETs, despite the demonstration through in vivo imaging and ICH of the expression of the target. In vitro experiments confirmed that TETs induce stromal cells to shift FN production to the ED-B subtype, likely representing a favorable hallmark for tumor progression and metastasis. Overall, our results suggested that many factors contribute to R-RIT inefficacy. Further research aimed to clarify the subcellular and molecular mechanisms underlying tumors’ phenotypes and treatment resistance will improve therapeutic protocols and patients’ outcome.

## Figures and Tables

**Figure 1 cancers-14-02592-f001:**
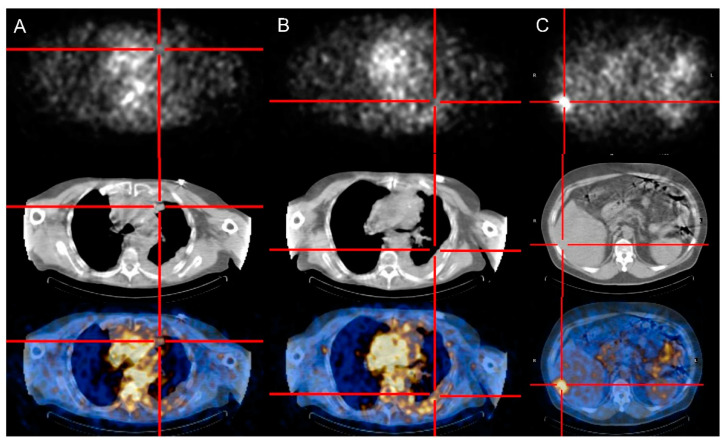
Patterns of Radretumab uptake at diagnostic phase. SPECT/CT images obtained 24 h post injection show absent (**A**), faint (**B**), and intense (**C**) Radretumab uptake, respectively.

**Figure 2 cancers-14-02592-f002:**
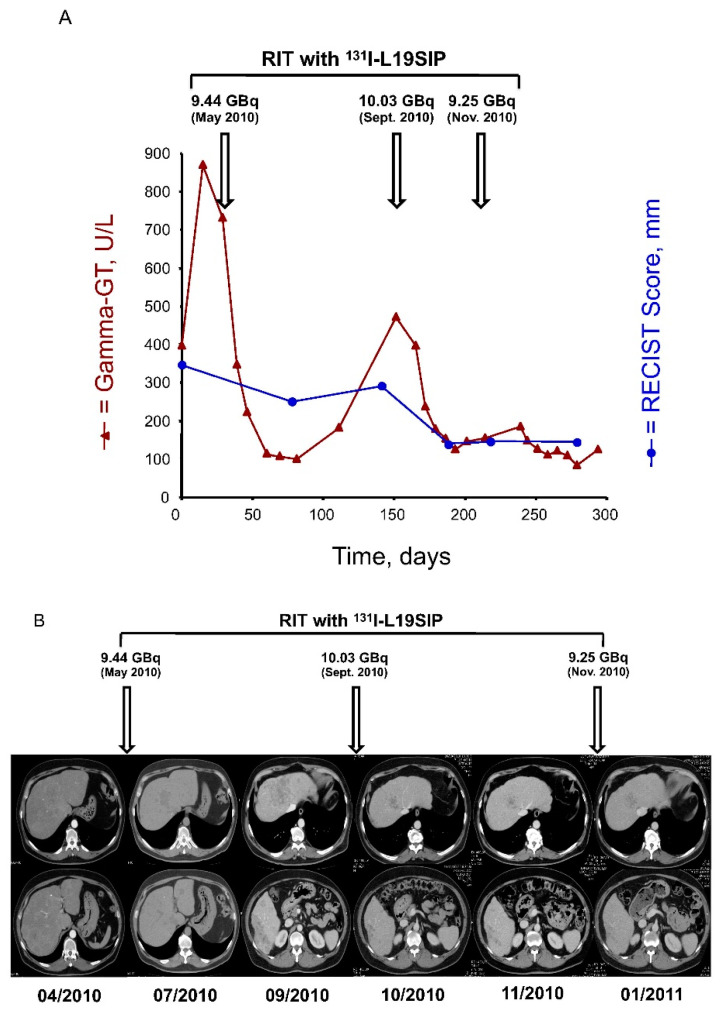
(**A**) Time course of changes in RECIST score (sum of maximum diameters of each target tumor lesion, expressed in mm) and in serum values of Gamma-GT (in U/L) over the period April 2010–February 2011. Over this period, three subsequent courses of R-RIT were administered, as indicated by arrows. The plots show objective responses after each R-RIT both in the serum marker of disease burden and in the RECIST score (−27.7% after R-RIT of 05/2010; −51.2% after R-RIT of 09/2010; disease remained stable after R-RIT of 11/2010). Each reduction in the Gamma-GT serum levels corresponded to considerable improvement of clinical symptoms (mostly nausea and vomiting). (**B**) CT slices showed the liver lesions of the VIII segment (top line) and the VI segment (bottom line) over the period April 2010–January 2011. Over this period, three subsequent courses of R-RIT were administered, as indicated by arrows.

**Figure 3 cancers-14-02592-f003:**
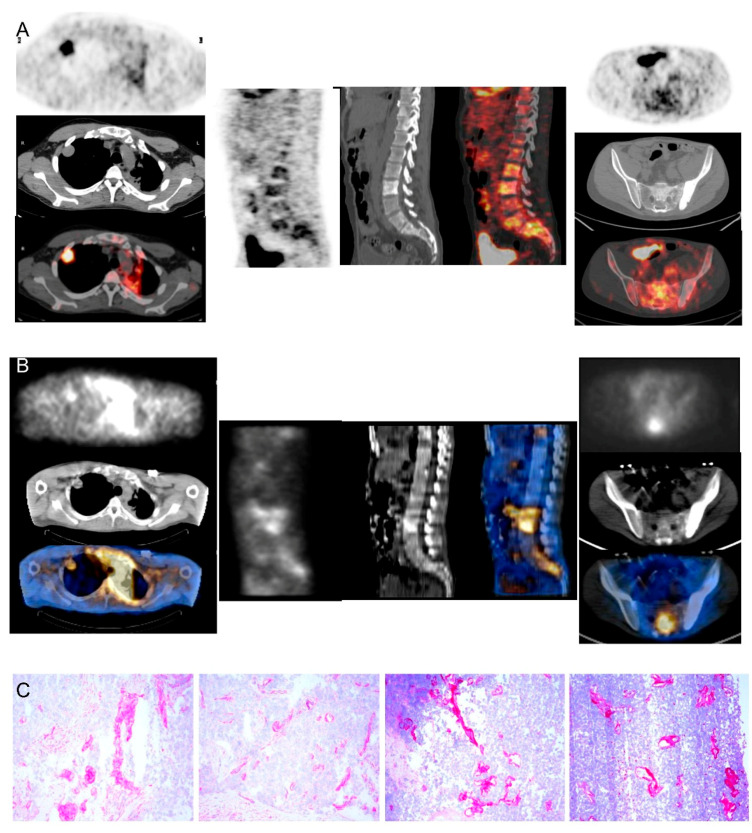
Comparison between [^18^F]FDG-PET/CT, Radretumab SPECT/CT, and immunohistochemical analysis in a thymoma B3 patient (#5). Pre-treatment PET/CT (**A**) shows high [^18^F]FDG uptake in a nodule in the right lung (left panels) and diffuse uptake in the spine (middle and right panels). SPECT/CT (**B**) outperforms PET/CT findings, showing highly specific Radretumab uptake in right lung (left panels) and spine (middle and right panels). Immunohistochemical analysis (**C**) shows intense ED-B expression in tissue samples, confirming SPECT/CT findings.

**Figure 4 cancers-14-02592-f004:**
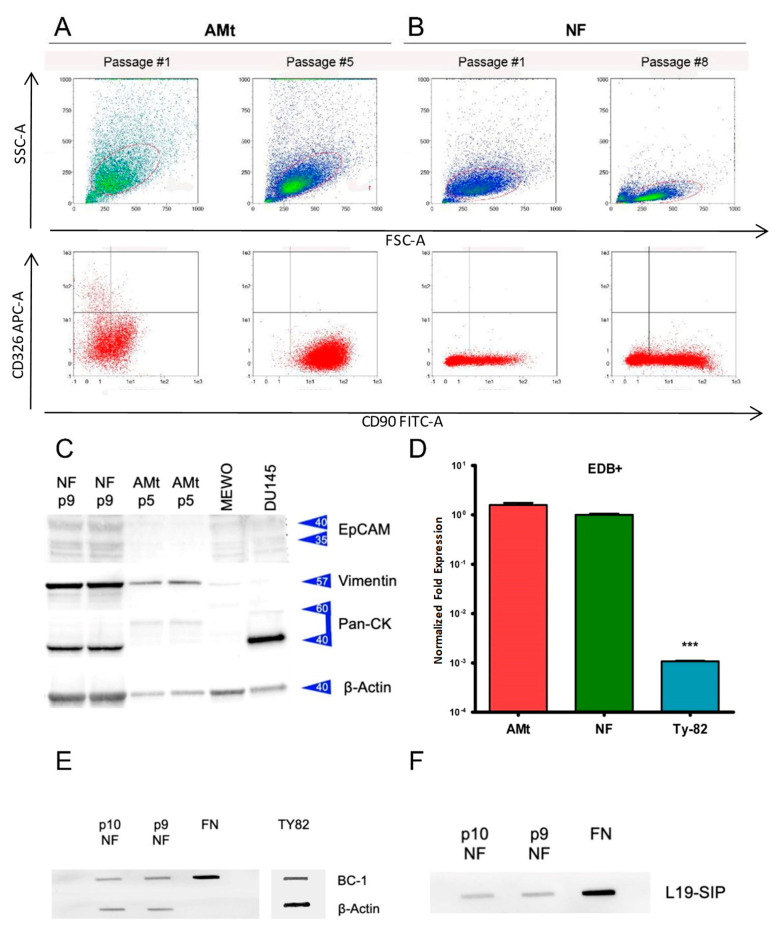
Immunophenotyping of primary cell culture in AMt (**A**) and NF (**B**) at the first and following passages. Flow cytometry shows the expression of thymocyte-mesenchymal (CD90^+^) and epithelial (CD326^+^) markers. CD326^+^ cells disappear at passage 5 in Amt, whereas they are absent in NF. (**C**) Western blot evaluating the expression of EpCAM, Vimentin, Pan cytokeratins (Pan-CK), and Beta acting as loading control in NF and AMt at passage 9 and 5, respectively. MEWO and DU145 cell lines were included as control. (**D**) Fold change expression of ED-B fibronectin in AMt and NF primary culture and in TY82 thymic carcinoma cell line (*** *p* < 0.001). The difference in expression was significative between primary cultures and TY82. Slot-blot assay of ED-B expression in NF cells at passage 9 and 10 was revealed by BC-1 monoclonal antibody (**E**) and L19-SIP (**F**).

**Table 1 cancers-14-02592-t001:** Main characteristics and results for each patient.

Pt	Age	Sex	Histology	ED-B FN Expression Visual Analysis	T/Non-T Ratio	AE/SAE Diagnostic	R-RIT (GBq)	AE/SAE R-RIT	Clinical Response	Radiological Response		
										Best Response	RECIST Score	Duration
#1	54	M	Thymic carcinoma	High	10.3 *	G3 increase creatinine	3.33	Worsening of pain and lymph edema	Worsening of clinical condition	NA		
#2	46	M	B3	High	78.6 *	Worsening of pre-existing asthenia associated with articular pain	5.55	Hypothyroidism		SD	−5%	5 mo
							5.55			PD	+68%	--
#3	64	M	Thymic carcinoma	Absent	1.2 *		NA					
#4	44	M	NA	Faint	6.4 *		NA					
#5	32	M	B3	High			5.18			SD		
							5.55			PD		
#6	53	M	Thymic carcinoma	High	54.4 ^		8.14	None		SD	−19%	3 mo
#7	53	F	B2	High	87.5 ^	G2 hypersensitivity to perchlorate potassium	NA					
#8	54	F	B2	High	64.7 ^		9.25	G4 lymphopenia (19 days after RIT, recovered) G3 thrombocytopenia (26 days after R-RIT, recovered) G2 anemia (26 days after R-RIT, recovered)		SD	+2%	5 mo
#9	54	M	Thymic carcinoma	High	8.1 ^	None	9.44	Iatrogenic hypothyroidism G2 abdominal cutaneous erythema (from day 30 to day 45)	Improvement of hepatic functionality indexes, nausea and vomiting	SD	−27.7%	3 mo
							10.03	G 3 lymphopenia (from day 29 to day 43)		PR	−51.2%	2 mo
							9.25	G3 leukopenia (from day 15 to day 34)		SD	−2%	2 mo
#10	49	M	Thymic carcinoma	Faint	1.6 ^	G2 dyspnea	NA					
#11	50	M	AB (re-classified as B3 in primary and B2 in metastatic lesions after biopsy revision)				2.96	G1 thrombocytopenia (from day 21 day 35)Hypothyroidism		PD		

M: male, F: female; G: grade; NA: not applicable; SD: stable disease; PR: partial response; PD: progressive disease; T/non-T ratio: target/non-target ratio; AE/SAE: adverse event/serious adverse event; R-RIT: Radretumab radioimmunotherapy; mo: months; ED-B FN: extra-domain B of fibronectin. * Target/non-target ratio defined as adsorbed dose in the target lesion/adsorbed dose in bone marrow. Eligibility for R-RIT ≥ 10. ^ Target/non-target ratio defined as adsorbed dose in the target lesion/adsorbed dose in muscle. Eligibility for R-RIT ≥ 4.

## Data Availability

Paola Anna Erba had full access to all the data in the study and takes responsibility for the integrity of the data and the accuracy of the data analysis. Raw data are available on specific request to the corresponding author.

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
