# Peer review of "ED-B-Containing Isoform of Fibronectin in Tumor Microenvironment of Thymomas: A Target for a Theragnostic Approach"

_cancers, 2022, doi:10.3390/cancers14112592_

Round 1
Reviewer 1 Report
The authors of the presented manuscript aimed to exploit tissue-specific differences of fibronectin (FN) expression in thymic epithelial tumors (TETs), present interesting clinical and limited molecular (in vitro) data, and conclude that stromal cells in thymoma are shifting their FN production to the extra-domain B subtype.
While the clinical data including immunohistochemistry is very interesting and relevant clearly showing the importance of imaging, the adjacent in vitro data is not very convincing and does not fully support the conclusion of R-RIT inefficacy being unlikely related to target expression in TET and do not provide any other supportive data for the actual reason instead.
Major issues:
- The purpose and the resulting meanings of the supporting in vitro experiments are not completely comprehensible because basic explanations are missing. I would suggest including more explanations for why the experiments were performed and what the results mean/depict.
- Protein loading for the Western Blot analysis was not equal for the samples as seen by the beta-actin staining. Thus, it is hard to compare the different samples. A semi-quantification, using e.g. ImageJ, would be advisory.
- The mentioned morphological properties of the cells should be shown as microscopic phase contrast images.
- A comparison of primary cells (i.e. AMt, NF) with a cell line (i.e. Ty82) is always critical and should be avoided. A better control for ED-B expression would be primary epithelial cells. This analysis should be repeated if possible.
- The major drawback of the presented study is that there are only speculations and no supportive data about other possible reasons for R-RIT inefficiency despite proper patient selection based on the imaging of target expression.
Minor issues:
- The language and grammar needs to be corrected. Some sentences are more short notes than full sentences, which results in sometimes-unclear statements.
- The labeling of the graphs in Figure 2 and Figure 5 is too small to read and should be modified.
- A discussion of the finding in terms of novel therapeutic options should be further discussed, e.g. including ED-B directed CART-cells (Wagner et al, Cancer Immunol Res 2021, 9 (3)).
Author Response
1. The purpose and the resulting meanings of the supporting in vitro experiments are not completely comprehensible because basic explanations are missing. I would suggest including more explanations for why the experiments were performed and what the results mean/depict.
We thank the Reviewer for this suggestion. In the revised version of the manuscript we detailed the purpose and significance of in vitro experiments.
2. Protein loading for the Western Blot analysis was not equal for the samples as seen by the beta-actin staining. Thus, it is hard to compare the different samples. A semi-quantification, using e.g. ImageJ, would be advisory.
We thank the reviewer for this comment. Accordingly, we added in the Supplementary Figures beta-actin staining and semi-quantitative analyses.
3. The mentioned morphological properties of the cells should be shown as microscopic phase contrast images.
We added required images as Supplementary Figure.
4. A comparison of primary cells (i.e. AMt, NF) with a cell line (i.e. Ty82) is always critical and should be avoided. A better control for ED-B expression would be primary epithelial cells. This analysis should be repeated if possible.
We thank the reviewer for this comment. Unfortunately, we were not able to fulfill this request for technical reasons. The primary cultures of thymic tumors are a mixture of stromal and epithelial cells. We try to isolate epithelial cells using EPCAM coated beads, but the best enrichment is around 30% of tumor cells (data not shown in this article). The expression of ED-B fibronectin using the available antibodies by western blot will be a mixed expression of all the cell populations. The only chance to obtain a measure of fibronectin expression in cancer cells is to use a thymic tumor cell line. The only available for us was TY82.
5. The major drawback of the presented study is that there are only speculations and no supportive data about other possible reasons for R-RIT inefficiency despite proper patient selection based on the imaging of target expression.
We thank the reviewer for this comment, and we modified accordingly the discussion.
Minor issues:
1. The language and grammar needs to be corrected. Some sentences are more short notes than full sentences, which results in sometimes-unclear statements.
We thank the reviewer for this comment. Typos and English language were carefully revised.
2. The labeling of the graphs in Figure 2 and Figure 5 is too small to read and should be modified.
We modified labeling of the above mentioned Figures as requested.
3. A discussion of the finding in terms of novel therapeutic options should be further discussed, e.g. including ED-B directed CART-cells (Wagner et al, Cancer Immunol Res 2021, 9 (3)).
We thank the reviewer for this comment. We added this novel therapeutic option in the discussion.

Reviewer 2 Report
The major weakness of the study is that it is not clearly presented. Especially the simple summary (lines 21 to 25), as well as the abstract need extensive editing as they are quite confusing. Moreover, abbreviations should be explained the first time mentioned, in order to be more understandable.
Another issue is the significance. The ED-B fibronectin was shown to be useful as biomarker for targeted therapy in cancer, and ways to visualise ED-B in tumor site (PET, IHC, WB, ELISA, qPCR) have also been previously studied. Thus, the new finding of this study is that thymic epithelial tumors induce ED-B FN production?
Furthermore, grammar and spelling need editing.
Approval number of the study should be included in Methods.
In Table 1, a description should replace the template text.
Author Response
1) The major weakness of the study is that it is not clearly presented. Especially the simple summary (lines 21 to 25), as well as the abstract need extensive editing as they are quite confusing. Moreover, abbreviations should be explained the first time mentioned, in order to be more understandable.
We thank the reviewer for these comments, and accordingly we revised the manuscript.
2)Another issue is the significance. The ED-B fibronectin was shown to be useful as biomarker for targeted therapy in cancer, and ways to visualise ED-B in tumor site (PET, IHC, WB, ELISA, qPCR) have also been previously studied. Thus, the new finding of this study is that thymic epithelial tumors induce ED-B FN production?
We thank the reviewer for this constructive comment. We modified the discussion providing a better explanation of significance of our results and adding literature data to support our speculations.
3)Furthermore, grammar and spelling need editing.
We thank the reviewer for this comment. Typos and English language were carefully revised.
4)Approval number of the study should be included in Methods.
We provided this detail.
5)In Table 1, a description should replace the template text.
We thank the reviewer for this comment and we modified accordingly Table 1.

Round 2
Reviewer 1 Report
All critics have been adressed by the authors. No further comments.
Reviewer 2 Report
No further comments.